# Fourier transform spectrometer on silicon with thermo-optic non-linearity and dispersion correction

Mario C.M.M. Souza[1], Andrew Grieco[2], Newton C. Frateschi[1] & Yeshaiahu Fainman[2]

Miniaturized integrated spectrometers will have unprecedented impact on applications ranging from unmanned aerial vehicles to mobile phones, and silicon photonics promises to deliver compact, cost-effective devices. Mirroring its ubiquitous free-space counterpart, a silicon photonics-based Fourier transform spectrometer (Si-FTS) can bring broadband operation and fine resolution to the chip scale. Here we present the modeling and experimental demonstration of a thermally tuned Si-FTS accounting for dispersion, thermo-optic non-linearity, and thermal expansion. We show how these effects modify the relation between the spectrum and interferogram of a light source and we develop a quantitative correction procedure through calibration with a tunable laser. We retrieve a broadband spectrum (7 THz around 193.4 THz with 0.38-THz resolution consuming 2.5 W per heater) and demonstrate the Si-FTS resilience to fabrication variations—a major advantage for large-scale manufacturing. Providing design flexibility and robustness, the Si-FTS is poised to become a fundamental building block for on-chip spectroscopy.

[1] "Gleb Wataghin" Physics Institute, University of Campinas, Campinas, SP 13083-970, Brazil. [2] Department of Electrical and Computer Engineering, University of California, San Diego, 9500 Gilman Drive, La Jolla, San Diego CA 92023, USA. Correspondence and requests for materials should be addressed to Y.F. (email: fainman@ece.ucsd.edu)

In recent years, significant efforts have been directed toward the realization of miniaturized optical spectrometers for in situ spectral analysis in numerous areas of science and technology[1–17]. The widespread use of optical spectroscopy from remote sensing[4,7] and planetary sciences[13] to medical research[18] and pharmaceutical processes[19] strongly relies on the large absorption and/or reflection cross-sections of many compounds in the near-infrared and mid-infrared range.

Silicon-based devices operate in such range and the substantial progress in integrated silicon photonics design and fabrication can therefore be leveraged to develop miniaturized spectrometers for mobile platforms. Typical silicon-on-insulator (SOI) waveguides can operate in the 1.1–4 µm wavelength range, limited by the silicon band edge at short wavelengths and by the oxide absorption at long wavelengths. The incorporation of additional complementary metal-oxide-superconductor (CMOS)-compatible materials to the mainstream fabrication process, including silicon nitride (SiN)[4,20] for short wavelengths and germanium-on-silicon (Ge-on-Si)[4,21,22] for long wavelengths, promises to significantly extend this window of operation. Moreover, the possibility of monolithic integration of spectrometers and photodetectors is a valuable advantage of photonic integration, promising high signal-to-noise ratio and increased sensitivity. Adding heterogeneously integrated light sources[23–25], all the optical components required for a fully functional spectrometer can be realized in a single chip. Finally, the access to multi-project-wafer services through silicon photonics foundries provides a cost-effective path to developing robust high-performance devices[4,26–28].

A large variety of integrated photonic spectrometer designs has been recently investigated. These include dispersive devices such as arrayed waveguide gratings[4,5] and cavity-enhanced spectrometers[6], spatial heterodyne spectrometers (SHS)[7–11] based on arrays of interferometers, and stationary wave-integrated Fourier transform spectrometers (SWIFTS)[12–14]. In the latter, a spatial—rather than temporal—interferogram is formed by the standing wave pattern generated by the interference of two counter-propagating beams inside a waveguide. Designs based on the traditional Fourier transform spectrometer (FTS)[29], in which a temporal interferogram is measured varying the optical path between the arms of an interferometer, have also been investigated using micro-electro-mechanical systems[15,16] and lithium-niobate planar photonic circuits[17].

Investigations of the traditional FTS design in the silicon photonics platform have been, however, surprisingly scarce[30]. Whereas FT-based spectrometers such as SHS are suitable for high-resolution, narrow-band applications, the traditional FTS is a promising candidate to address moderate resolution, broadband applications. Considering the requirements for a silicon photonics-based FTS (Si-FTS), some challenges can be identified. First, the optical path difference between the arms of the interferometer is achieved tuning the refractive index rather than changing the physical length, thus an index tuning mechanism capable of large index changes must be used for high spectral resolution. Fortunately, the thermo-optic effect can deliver large index changes of more than $10^{-2}$ for temperature differences around 100 K. For large temperatures, however, thermo-optic non-linearity and thermal expansion of the waveguide become important. Second, silicon waveguides are highly dispersive. As a consequence, the thermo-optic effect will also present strong dispersion. This translates to each optical frequency effectively experiencing a different change in optical path.

In this article, we demonstrate the implementation of a Si-FTS on the SOI platform with integrated microheaters. We show that the issues related to thermo-optic non-linearity, thermal expansion, and dispersion can be properly understood and incorporated in a simple manner. We derive a FT relation between the power spectral density (PSD) and the interferogram with modified optical frequency and arm delay accounting for these effects and we demonstrate a calibration procedure using a tunable laser source. Further, we demonstrate the retrieval of a 7-THz-wide light source around 193.4 THz with spectral resolution of 0.38 THz (12.7 cm$^{-1}$, 3.05 nm) using a 1 mm$^2$ device with total electric power dissipation around 2.5 W per heater. The Si-FTS shows intrinsic resilience to fabrication variations that allows scalability of its resolution and power consumption performance, enabling robust and versatile portable spectrometers.

## Results

**Experimental device**. The device consists of a standard Mach-Zehnder Interferometer (MZI) integrated with metal microheaters fabricated in full compatibility with standard silicon photonics foundry processes (Fig. 1). The external light is butt-coupled into and out of the chip using inverse tapers and adiabatically transitions to the highly confined quasi-TE mode of the

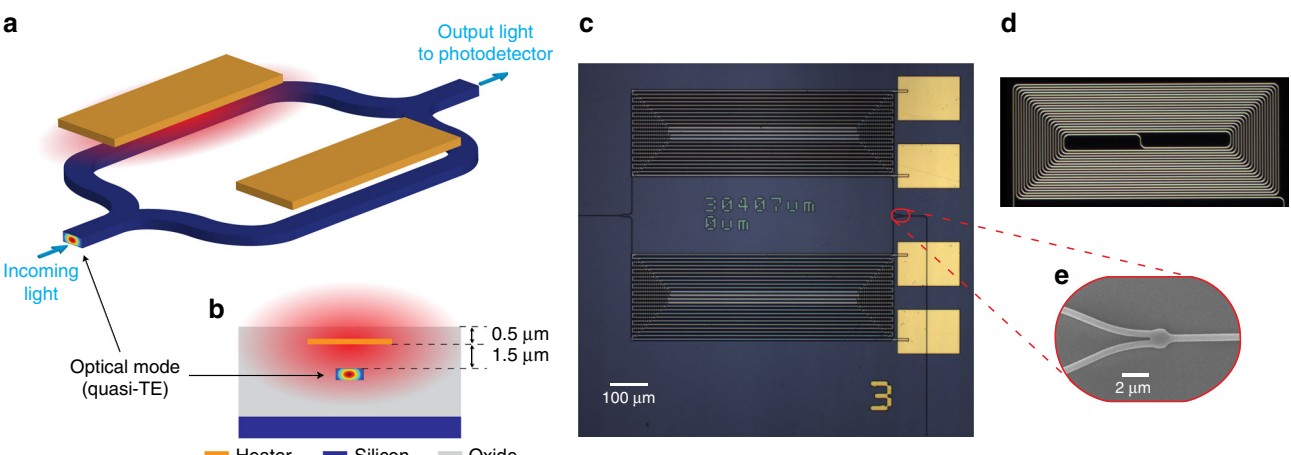

**Fig. 1** On-chip Fourier transform spectrometer. **a** Schematic of a MZI with integrated metal microheaters on silicon-on-insulator (SOI) platform. **b** Device cross-section illustrating the quasi-TE mode (energy density) of the strip silicon waveguide and the heated area (light red) when current flows through the microheater. **c** Optical micrography of the experimental device with a total footprint of 1 mm$^2$ (see fabrication details in the Methods section). **d** Dark field optical micrography of the MZI arm underneath the heater trails. **e** SEM image of the broadband power splitter/combiner

access strip waveguide before splitting in the two arms of the interferometer and subsequently recombining into the output waveguide through broadband y-branch couplers (Fig. 1e)[31]. The output light is coupled out of chip directly into a photodetector. Each arm of the MZI consists of a spiral (Fig. 1d) with total length of 30.407 mm and is covered by independently actuated nichrome microheaters. The propagation losses of the waveguides are estimated to be around 2 dB cm$^{-1}$. The total device footprint is 1 mm$^2$.

**Spectrometer modeling**. In this section, we present the main results that allow to define a modified FT relation between the varying optical power at the output of the MZI, $I$, and the PSD of the incoming light, $PSD(\nu)$. A detailed discussion is provided in Supplementary Note 2.

The operation of the Si-FTS includes a simple data acquisition step consisting of measuring the output power as a function of the phase difference $\Delta\phi$ between the two arms of the MZI. The $\Delta\phi$-dependent term is given by

$$I(\Delta\phi) = \int_{-\infty}^{+\infty} T(\nu)\,PSD(\nu)e^{j\Delta\phi(\nu)}\,d\nu, \qquad (1)$$

where $\nu$ is the optical frequency and $T(\nu)$ is the transfer function of the MZI—ideally 1. The phase difference is

$$\Delta\phi(\nu) = \frac{2\pi\nu}{c}\left[n_{eff,1}(\nu)L_1 - n_{eff,2}(\nu)L_2\right], \qquad (2)$$

where $c$ is the speed of light, $n_{eff,i}$ and $L_i$ are the effective index and the total length of arm $i$.

The discussion is facilitated by first considering the response of an idealized device. In this case $T(\nu)=1$, the two arms are identical with length $L$, the effective indices are identical and dispersionless, $n_{eff,i}(\nu) \equiv n_{eff}$, and the effective index change due to temperature change $\Delta T$ depends only on a linear thermo-optic coefficient (TOC) $\partial_T n$, such that $\Delta n_{eff} = \partial_T n\Delta T$. We use a contracted notation for partial derivatives, $\frac{\partial n_{eff}}{\partial x} \equiv \partial_x n$. The time delay between the arms of the MZI is defined as $\tau = \frac{L}{c}\partial_T n\Delta T$ and the phase difference is simply

$$\Delta\phi(\nu) = 2\pi\nu\tau. \qquad (3)$$

The phase difference in the form $2\pi \times$ frequency $\times$ delay establishes a direct FT relation between $I(\tau)$ and $PSD(\nu)$, with the conjugate variables $\nu$ and $\tau$,

$$I(\tau) = \int_{-\infty}^{+\infty} PSD(\nu)e^{j2\pi\nu\tau}d\nu = \mathcal{F}[PSD(\nu)], \qquad (4)$$

where $\mathcal{F}[]$ denotes the Fourier transform. Thus, $PSD(\nu)$ can be directly obtained from the inverse FT (IFT) of the interferogram,

$$PSD(\nu) = \int_{-\infty}^{+\infty} I(\tau)e^{-j2\pi\nu\tau}\,d\tau = \mathcal{F}^{-1}[I(\tau)]. \qquad (5)$$

In practice, the Si-FTS with thermal tuning includes other effects that must be taken into account. First, the strong mode dispersion of silicon waveguides causes significant frequency dependence on the effective index. Second, a large temperature excursion is required to achieve large phase imbalances and the non-linearity of the thermo-optic response must be considered. The large temperature excursion also induces changes in the arm length ($\Delta L$) due to thermal expansion. Finally, chip-scale variability[20,26] and fabrication imperfections often introduce small differences between the two arms of the MZI, identical by

design. Such variations may affect the arm length ($\delta L$) as well as the effective index ($\delta n(\nu)$). Assuming the heater on top of arm 1 ($H_1$) is actuated and including the deviations from the designed parameters in arm 2, the effective indices and arm lengths are

$$\begin{aligned} n_{eff,1}(\nu,\Delta T) &= n_{eff}(\nu) + \Delta n_{eff}(\nu,\Delta T), \\ n_{eff,2}(\nu) &= n_{eff}(\nu) + \delta n(\nu), \\ L_1(\Delta T) &= L + \Delta L(\Delta T), \\ L_2 &= L + \delta L \end{aligned} \qquad (6)$$

The expressions for $n_{eff}(\nu)$, $\Delta n_{eff}(\nu, \Delta T)$, $\delta n(\nu)$, and $\Delta L(\Delta T)$ are presented in Supplementary Note 2 and contain high-order terms in $\Delta T$ and/or $\Delta\nu$, with $\Delta\nu = \nu - \nu_0$.

The phase difference of the Si-FTS can be written in the form $2\pi \times$ frequency $\times$ delay, similar to Eq. 3 but with modified frequency and delay terms. Substituting Eq. 6 in Eq. 2 yields $\Delta\phi$ with contributions in $\Delta\nu^i\Delta T^j$ for $i$ from 0 to 4 and for $j$ from 0 to 5. The simplification of the resulting expression depends on the dispersion and thermo-optical properties of the waveguide as well as on the maximum temperature excursion and on the bandwidth of the light source. Using the properties of our 250-by-550 nm$^2$ SOI waveguides (Supplementary Table 1) with temperature excursion of <100 K and optical bandwidth of a few tens of terahertz, we show that $\Delta\phi$ can be simplified to

$$\Delta\phi \approx \varphi(\nu) + 2\pi u\mathcal{T}. \qquad (7)$$

The first term $\varphi(\nu)$ depends only on $\nu$ (Supplementary Eq. 19) and therefore does not contribute to the kernel of the FT. It contributes to shifting and distorting the interferogram, but it has no influence on the retrieved PSD as will be shown. Thus, it is not discussed in detail here.

The second term, $2\pi u\mathcal{T}$ is similar to Eq. 3, but with modified optical frequency and time delay

$$\begin{aligned} u &= \Delta\nu(1 + \xi_1) + \nu_0 \\ \mathcal{T} &= \tau + \gamma_2\tau^2. \end{aligned} \qquad (8)$$

$u$ represents the broadening of the original optical frequency $\nu$ around $\nu_0$ by a factor $1 + \xi_1$. In our case, $\xi_1$ is dominated by $\partial_{\nu,T}n$. $\mathcal{T}$ represents a correction of the original optical delay $\tau$ that includes the non-linear contribution $\gamma_2\tau^2$, with $\gamma_2$ dominated by $\partial_{T^2}n$. $\tau$ in turn is related to $\Delta T$ by $\tau = \eta_1\Delta T$, with $\eta_1$ dominated by $\partial_T n$. The expressions for $\xi_1$, $\gamma_2$, and $\eta_1$ are presented in Supplementary Note 2.

Substituting Eq. 7 in Eq. 1 and changing the integration variable from $\nu$ to $u$, the interferogram and the PSD are finally related through a FT in the modified conjugate variables $u$ and $\mathcal{T}$, denoted by $\mathcal{F}[]$,

$$I(\mathcal{T}) = \frac{1}{1 + \xi_1}\mathcal{F}\left[e^{j\varphi(u)}T(u)\,PSD(u)\right]. \qquad (9)$$

Neglecting the constant term $(1 + \xi_1)^{-1}$ multiplying the FT, the PSD is then retrieved from the absolute value of the IFT of the interferogram, normalized by the MZI transfer function $T(u)$,

$$PSD(u) = \frac{\left|\mathcal{F}^{-1}[I(\mathcal{T})]\right|}{T(u)}. \qquad (10)$$

Finally, the frequency axis must be transformed back to the original optical frequency $\nu$,

$$PSD(u) \xrightarrow{\nu = \frac{u-\nu_0}{1+\xi_1}+\nu_0} PSD(\nu). \qquad (11)$$

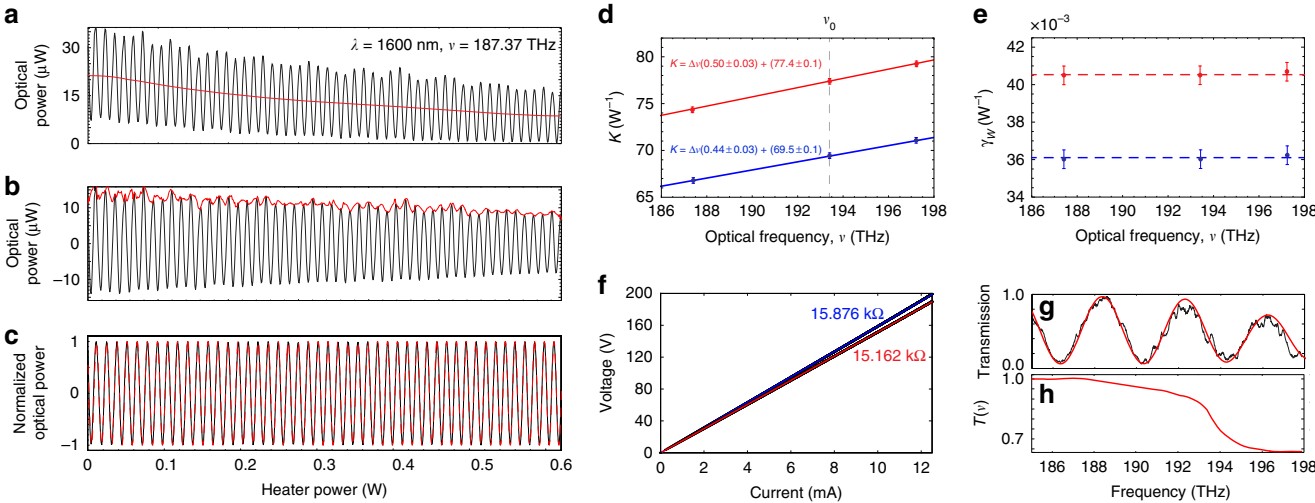

**Fig. 2** Si-FTS calibration using a tunable laser source. Measurements are performed in the C-band. **a–c** Interferogram at 183.37 THz (1600 nm) as a function of dissipated power in heater $H_2$. The mean power (red trace in **a**) and the envelope (red trace in **b**) are subtracted to obtain the curve in **c**, fitted (dashed-red trace) using Eq. 13. The envelope in **b** is the absolute value of the interferogram's Hilbert transform. **d–f** Data in blue and red are related to heaters $H_1$ and $H_2$, respectively. **d** Parameter $K(\nu)$ obtained from the non-linear fit, adjusted according to Eq. 14. Error bars are s.d. (95% confidence level). **e** Non-linear parameters $\gamma_{W,1} = (36.2 \pm 0.3) \times 10^{-3}\,\mathrm{W}^{-1}$ and $\gamma_{W,2} = (40.5 \pm 0.3) \times 10^{-3}\,\mathrm{W}^{-1}$ obtained from the non-linear fit. Error bars are s.d. (95% confidence level). **f** Current vs voltage (IV) response of both heaters and calculated electric resistance. **g** Experimental (black trace) and calculated (red trace) transmission spectrum of the MZI at non-zero optical delay (0.172 ps). The calculated transmission is obtained using Eq. 15 to extract the MZI transfer function $T(\nu)$ shown in **h**

**Spectrometer calibration with tunable laser.** As in free space, the Si-FTS must be calibrated to provide good absolute frequency accuracy. In addition, parameters $\xi_1$, $\gamma_2$, and $T(\nu)$ should also be ideally determined in a calibration step such that the transformations $u \rightarrow \nu$, $\tau \rightarrow \mathcal{T}$, and the re-normalization of Eq. 10 can be properly performed in later use. A calibration process realized with a narrow linewidth tunable laser source allows to address all these requirements.

First, the calibration of the absolute optical frequency, $\xi_1$ and $\gamma_2$ is achieved measuring the interferogram of the laser source at different laser frequencies (at least three) in the spectral region of interest. Calibrating the absolute optical frequency reduces to determining $\kappa_\tau$ that connects the electric power dissipated in the heater with the resulting arm delay, $\tau = \kappa_\tau W$. The interferogram of a laser source at frequency $\nu$ is, according to Eq. 9,

$$I(\mathcal{T}) = A\cos(2\pi u\mathcal{T} + \varphi_\nu), \qquad (12)$$

where $A$ and $\varphi_\nu$ are constant amplitude and phase. Using this relation and Eq. 8, the interferogram can be written as a function of the electric power and the original laser frequency as

$$I(W, \nu) = A\cos[2\pi K(\nu)\,W(1 + \gamma_W W) + \varphi_\nu] \qquad (13)$$

with

$$\begin{aligned} K(\nu) &= \kappa_\tau(1 + \xi_1)\,\Delta\nu + \kappa_\tau\nu_0 \\ \gamma_W &= \kappa_\tau\gamma_2. \end{aligned} \qquad (14)$$

$K(\nu)$ and $\gamma_W$ can be determined for each heater ($H_1$ and $H_2$) curve-fitting the experimental interferograms using a cosine with non-linear argument. Following, the linear fit of $K(\nu)$ with $\nu_0 = 193.414$ THz allows to determine $\kappa_\tau$ for each heater and $\xi_1$. Finally, using $\kappa_\tau$, we obtain $\gamma_2 = \gamma_W/\kappa_\tau$.

The calibration results using the laser interferograms are presented in Fig. 2a–e and the extracted parameters are summarized in Table 1. The interferogram for the laser frequency 187.37 THz (1600 nm) with heater $H_2$ actuated and its curve-fit

| Table 1 Parameters obtained from calibration | |
|---|---|
| **Parameter** | **Value** |
| $\kappa_{\tau,1}$ ($10^{-3}$ ps $\times$ W$^{-1}$) | $359 \pm 1$ |
| $\kappa_{\tau,2}$ ($10^{-3}$ ps $\times$ W$^{-1}$) | $400 \pm 1$ |
| $\gamma_2$ ($10^{-3}$ ps$^{-1}$) | $101 \pm 1$ |
| $\xi_1$ ($10^{-2}$) | $23 \pm 1$ |

are shown in Fig. 2a–c as an example of the procedure realized for multiple frequencies and both heaters. The decrease in the mean optical power in Fig. 2a and the small features in the envelope of Fig. 2b are caused by slight misalignment and vibration of the input/output fibers as the heater power increases due to the thermal expansion of the chip. In a practical application, these features would not be present as the fiber would be permanently attached to the silicon chip. Combining the interferogram fit results for each heater, $K(\nu)$ follows a linear dependence with frequency (Fig. 2d) while $\gamma_W$ has a constant value (Fig. 2e), in agreement with the Eq. 14. The non-linear term $\gamma_2$ obtained separately for each heater has the same value within the experimental error, $(101 \pm 1) \times 10^{-3}$ ps$^{-1}$, while $\xi_1$ only slightly deviates for each heater, $0.22 \pm 0.02$ for $H_1$ and $0.24 \pm 0.01$ for $H_2$, yielding $0.23 \pm 0.02$.

The current vs voltage (IV) curves depicted in Fig. 2f confirm the thermo-optic origin of the measured non-linearity and allow to understand the difference between $\kappa_{\tau,1}$ and $\kappa_{\tau,2}$. The IV plots show a fairly linear behavior for both heaters, with resistances 15,876 k$\Omega$ for $H_1$ and 15,162 k$\Omega$ for $H_2$, and indicates that any non-linearity originating from the heaters is small compared to the non-linearity intrinsic to the thermo-optic effect. Moreover, the difference in electric resistance causes a difference in heater efficiency $k_T$ that explains the observed discrepancy in the measured $\kappa_\tau$'s. The heater efficiency is determined such that the temperature change at the waveguide level $\Delta T$ is related to the dissipated electric power $W$ by $\Delta T = k_T W$, thus $\kappa_{\tau,i} = \eta_1 k_{T,i}$. The

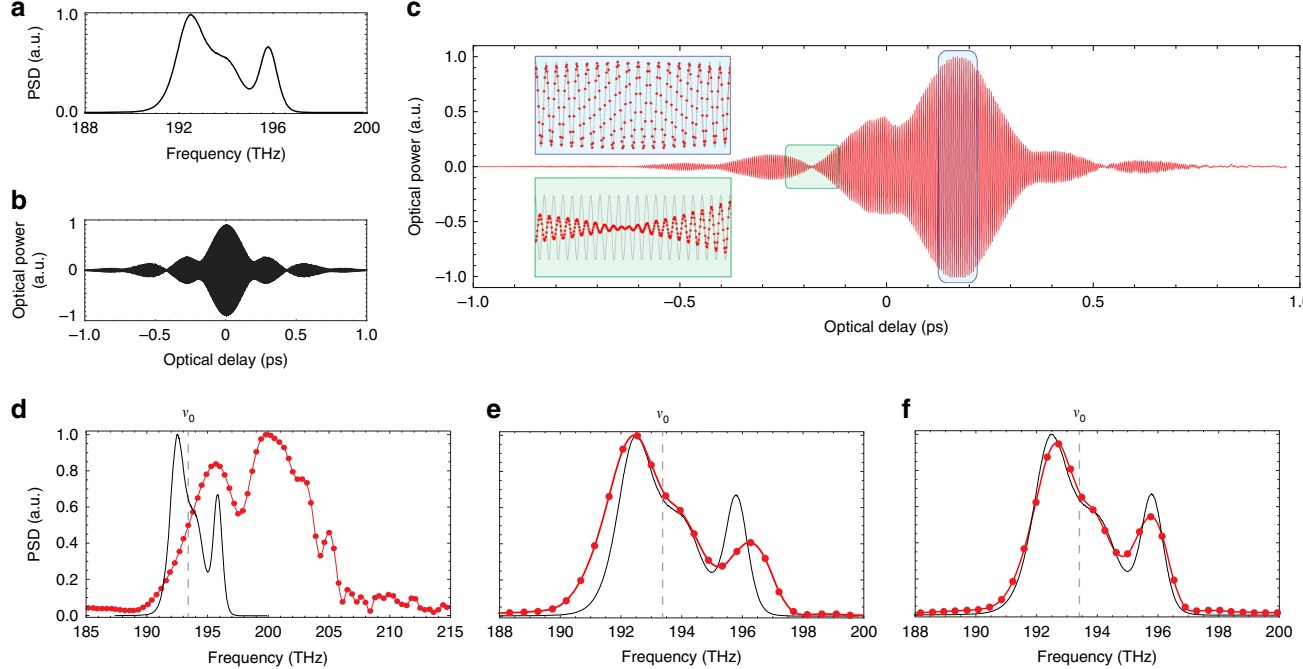

**Fig. 3** Broadband spectrum recovery with the Si-FTS. The parameters used to obtain these plots are summarized in Table 1. **a** ASE of a C-band EDFA used as the light source. **b** Theoretical interferogram of the ASE for an ideal (linear TOC, dispersionless, balanced) MZI. **c** Experimental interferogram, shifted to 0.172 ps and distorted due to differences between the two arms of the MZI. The optical delay axis corresponds to $\mathcal{T}$. The insets show a zoom-in of the interferogram at different optical delays superposed to a cosine (gray traces) at the ASE mean frequency $\nu_s = 193.44$ THz, highlighting: the oscillations at $\nu_s$ when the envelope varies slowly (blue-colored zoom-in); phase changes when the envelope varies rapidly (green-colored zoom-in). **d–f** Experimental (red) and reference (black) PSD at different conditions. The red points are the experimental data obtained from the IFT and the red line is a second-order interpolation curve. **d** No correction. **e** Corrected thermo-optic non-linearity, but no dispersion correction nor PSD re-normalization with $T(\nu)$. **f** All effects properly accounted for

agreement in $\gamma_2$ and $\xi_1$ obtained independently for each heater indicates the two arms of the MZI are fairly similar, so that $\eta_1$ can be considered the same. Using $\eta_1 \approx 1.94 \times 10^{-2}$ps K$^{-1}$ obtained from simulations (Supplementary Note 1), the heater efficiencies are estimated at $k_{T,1} \approx 18.5$ K W$^{-1}$ and $k_{T,2} \approx 20.6$ K W$^{-1}$.

The interferometer transfer function $T(\nu)$ is obtained from the transmission spectrum of the MZI, as depicted in Fig. 2g, h. The optical power at the output of the MZI is

$$I_{\text{out}}(\nu) = I_0(\nu) + T(\nu)\cos[\Delta\phi(\nu)]. \tag{15}$$

$T(\nu)$ is the envelope of the transmission oscillations and can be obtained by adjusting the experimental trace. It is recommended that the transmission spectrum be measured at a non-null phase difference $\Delta\phi$ so that $T(\nu)$ can be decoupled from the frequency dependence of the average power $I_0(\nu)$. In our case, the experimental transmission (black trace) represents the transmission spectrum of the passive device when none of the heaters are actuated. The oscillations have a free-spectral range of 3.96 THz around 193.414 THz (32.7 nm around 1550 nm) originated from slight differences between the two arms incorporated as $\delta L$ and $\delta n$($\nu$) as discussed in the previous section. The non-flat transmission of Fig. 2h is attributed to the non-ideal response of the power splitter/combiner for frequencies higher than 194 THz. In practice, the limited bandwidth of these components will dictate the bandwidth of the Si-FTS. For the specific y-junction design used here[31], operation over >25 THz (200 nm), excess loss lower than 0.3 dB and low reflection is expected.

**Broadband spectrum recovery.** The Si-FTS is validated by recovering the spectrum of the amplified spontaneous emission

(ASE) of a C-band erbium-doped fiber amplifier (EDFA). The ASE provides a good test spectrum in the telecom band, suitable for testing with the available equipment in our lab. Also, the broad features of the ASE spectrum are suitable for this demonstration given the limited resolution achieved here (0.38 THz). The reference ASE spectrum, measured with a tabletop optical spectrum analyzer, is shown in Fig. 3a, and its theoretical ideal interferogram for a dispersionless, perfectly balanced Si-FTS is depicted in Fig. 3b. The ASE presents two resolved peaks at 192.6 and 195.9 THz in addition to an unresolved peak at 194 THz, with total bandwidth around 7 THz (56 nm, 233.5 cm$^{-1}$).

The experimental interferogram is presented in Fig. 3c. The delay axis corresponds to the transformed delay $\mathcal{T}$ obtained using the parameters from Table 1. Positive delay corresponds to heater $H_1$, while negative delay corresponds to $H_2$. It spans $\Delta\mathcal{T} = 2.13$ ps, corresponding to maximum dissipated powers around 2.6 and 2.5 W in heaters $H_1$ and $H_2$ and maximum temperature excursions around 54 and 46 K at the waveguide level.

A comparison between experimental and ideal interferograms shows some noticeable distinctions. The zero delay (maximum envelope amplitude) is shifted to around 0.172 ps and the interferogram envelope is asymmetric. These effects result from the non-zero contribution of $\varphi(\nu)$ in Eq. 9, which carries the contribution of the difference between arms, $\delta L$ and $\delta n(\nu)$ (Supplementary Eq. 20). The zero-delay shift is caused by the first-order in $\nu$, while the envelope distortion is caused by higher-order terms.

The difference between arms that cause these observable changes in the interferogram correspond to typical variations expected from the silicon photonics process, rather than strong differences due to a non-ideal fabrication process. Considering

the difference in arm length $\delta L$ negligible, the zero delay is centered at

$$\mathcal{T}_0 \approx \frac{L}{c} \frac{\delta\left(n_{\text{eff}}|_{\nu_0}\right) + \nu_0 \delta(\partial_\nu n)}{1 + \xi_1}. \tag{16}$$

Using the known values for the other parameters, we estimate $\delta\left(n_{\text{eff}}|_{\nu_0}\right) + \nu_0 \delta(\partial_\nu n) \approx 2 \times 10^{-3}$. This value agrees with the expected order of magnitude for effective index fluctuations due to chip-scale variations in the silicon device layer thickness[20,26]. The same is true for the differences in high-order dispersion terms.

The interferogram oscillations are highlighted in the insets of Fig. 3c, where a cosine oscillation at the ASE mean frequency ($\nu_s = 193.44$ THz) is superposed (gray trace). In regions with a smooth envelope variation the interferogram oscillates at $\nu_s$ (blue-colored zoom-in), while an additional phase is introduced as the envelope varies more abruptly (green-colored zoom-in). The phase change due to envelope variations supports the interest of using a narrow laser source with almost flat envelope to calibrate the thermo-optic non-linearity, as the non-linear fit would be compromised by such phase change.

The PSD obtained from the experimental interferogram and the effects of thermo-optic non-linearity ($\gamma_2$), dispersion ($\xi_1$), and MZI transfer function ($T(\nu)$) are presented in Fig. 3d–f. The spectra were calculated using the mean value of the calibration parameters summarized in Table 1.

The PSD obtained directly form the as-measured interferogram—with the delay axis corresponding to $\tau$ and without performing any correction—is presented in Fig. 3d. The TOC non-linearity distorts, broadens, and shifts the PSD to higher frequencies as the interferogram oscillates faster with increasing delay.

After the optical delay axis of the interferogram is properly transformed to $\mathcal{T}$, the resulting PSD becomes very similar to the reference spectrum (Fig. 3e). Both resolved peaks are clearly identified and the unresolved peak is also present around 194 THz. However, since the spectrum has not been re-scaled to the original frequency $\nu$, it is broadened by the factor $1 + \xi_1$ around $\nu_0$. In addition, since it has not been re-normalized by $T(\nu)$, the high frequency peak appears attenuated relatively to the low frequency peak.

Failing to re-scale the frequency axis according to Eq. 11 introduces a frequency error $u - \nu = \xi_1 \Delta\nu$ that degrades the absolute frequency accuracy. In this demonstration, the effect is modest since the detuning of the peaks with respect to $\nu_0$ is small. The peak around 196 THz, for instance, is shifted by 0.5 THz, which is roughly half of its full-width at half maximum. Nonetheless, the effect can be very significant for frequencies further from $\nu_0$.

The PSD corrected for the thermo-optic non-linearity, dispersion, and the MZI transfer function reproduces satisfactorily well the reference spectrum (Fig. 3f). This spectrum was obtained performing only the aforementioned corrections, with no additional data processing such as zero-filling or apodization of the interferogram. The experimental spectral resolution of $\delta\nu = 0.38$ THz ($\delta\sigma = 12.7$ cm$^{-1}$, $\delta\lambda = 3.05$ nm) is comparable to other on-chip spectrometers aimed at broadband operation and it is sufficient for a large range of Raman and infra-red (IR) absorption spectroscopy applications[14,16].

## Discussion

The ultimate performance of the on-chip Si-FTS is quite promising considering recent advancements in silicon photonics design and fabrication. First, the window of operation for a given device will be dictated by the finite bandwidth of the waveguide

optical power couplers/splitters. Such components offering flat optical response over tens of terahertz[31–33] (hundreds of nanometers) and extremely low excess loss may allow Si-FTS operating over large bandwidths. Second, fine spectral resolution could be achieved using long low-loss silicon waveguides fabricated in tight footprints[28,34–36] combined with high temperature excursions endured by CMOS-compatible silicon devices[37]. Finally, the power efficiency can be significantly improved by applying suitable design changes. For instance, using Michelson interferometers[38] instead of MZIs can double the optical path in a given footprint, while introducing heat isolating structures can significantly increase heating efficiency[39,40].

In addition to high performance, a valuable advantage of the Si-FTS compared to other on-chip spectrometer approaches is its robustness to fabrication variations. Although the interferogram is strongly affected by the difference in effective index between the arms of the MZI (Fig. 3c), as previously discussed, the PSD remains unaffected (Fig. 3f). This result is expected from our model since $\varphi(u)$ is canceled calculating the PSD by taking the absolute value of the IFT (Eq. 10).

It is worth noticing that the presence of TOC non-linearity and dispersion have an upside in the Si-FTS performance. Ideally, one seeks to lower the resolution × dissipated power product, given by

$$\delta\nu \times W_{\text{total}} = \kappa_\tau^{-1}$$

in the case where the two effects are absent ($\kappa_\tau$ is assumed to be identical for the two heaters). In the presence of $\xi_1$ and $\gamma_2$, this product is modified to

$$\delta\nu \times W_{\text{total}} = \kappa_\tau^{-1} \left[(1 + \xi_1)(1 + \gamma_2 \kappa_\tau W_{\text{total}})\right]^{-1}$$

and is effectively reduced, resulting in decreased power dissipation for a given resolution or, conversely, lower $\delta\nu$ given a maximum power. In our case, the resolution of 0.38 THz is achieved dissipating a total power of 5.1 W, representing a 35% decrease in the power (6.9 W) that would be required to achieve this resolution if dispersion and thermo-optic non-linearity were absent.

In summary, we demonstrated the realization of a FTS with true time delay in silicon photonics. Considering the non-linearity of the thermo-optic effect as well as thermal expansion and dispersion, we derived simple corrections that effectively account for these effects and allow to use well established FT techniques to obtain accurate spectral responses. We showed how the Si-FTS can be calibrated using a tunable laser source and we demonstrated the successful recovery of a broadband spectrum, resilient to fabrication variations. Our discussion proposes a simple approach to tackle the hurdles of doing FT spectrometry using dispersive integrated platforms with high temperature excursions that can be readily applied to other device geometries and extended silicon photonics platforms such as SiN and Ge-on-Si, paving the way for robust, cost-effective, and versatile FT-based portable spectrometers.

## Methods

**Fabrication**. The Si-FTS was fabricated in $15 \times 15$ mm$^2$ dies from 250-nm-thick SOI wafers with 3 μm of buried oxide. All the fabrication steps correspond to standard CMOS-compatible processes. The waveguides (550 nm wide) and input/output taper regions (150 nm wide at the tip) were patterned using electron-beam (e-beam) lithography with the negative tone resist hydrogen silsequioxane (HSQ). The separation between adjacent waveguides in the spiral sections is 5 μm center-to-center and the bending radius is 10 μm. After patterning, the waveguides are dry etched, the HSQ is striped with a dip in buffered oxide etch solution and the waveguides are covered by 1.5 μm of plasma-enhanced chemical vapor deposition (PECVD)-deposited silicon oxide cladding. After the oxide deposition, the heaters are patterned on top of each arm using PMMA and the metals are deposited via sputtering in a liftoff process. The heaters consist of a serpentine nichrome (NiCr) trail of 6 μm-wide 280 nm-thick sections separated by 4 μm border-to-border and

totaling 17.33 mm in length. The NiCr heaters are terminated by $170 \times 120\ \mu m^2$ titanium-gold (Ti:Au, 5 nm:300 nm) pads. A second PECVD silicon oxide layer (500 nm-thick) is deposited on top of the heaters and $100 \times 100\ \mu m^2$ windows are opened on top of the Ti:Au pads to allow electric contact. After fabrication, the devices are diced and the input/output facets are polished to mitigate optical losses.

**Electrical and optical measurements**. The calibration described in the main text was performed using a tunable laser source (Agilent 81600B) operating between 1460 and 1630 nm with optical power around 0 dBm. The EDFA broadband source (Amonics) delivered around 15 dBm of total optical power in the spectral range between 1528 and 1564 nm and was connected to an in-line fiber polarizer. Light was coupled into and out of the device using polarization maintaining cleaved fibers, oriented to excite the quasi-TE mode of the SOI waveguide. The output fiber was directly connected to an InGaAs powermeter. The coupling fibers were positioned using precise translation stages (Thorlabs NanoMax) driven by piezoelectric controllers. The fibers made physical contact with the sample, but were not permanently attached to it. The sample was mounted on top of a temperature stabilized station (20 °C). The heaters were driven using a sourcemeter (Keithley 2400) allowing for simultaneous precise current drive (up to 2.1 A) and voltage monitoring (up to 210 V). The electric contact with the sample was done using microprobes with $50 \times 50\ \mu m^2$ tips connected to the heater pads. The total parasitic series resistance of the electric apparatus was <4 ohm. All the equipment was connected to a computer and controlled via Matlab routines. The interferograms were measured sweeping the electric current from zero to 2 A using a non-linear vector increasing with the square of the current such that the electric power vector would be linearly spaced. The interval between consecutive measurements was <0.5 s. Notice that a continuous current sweep was not employed only because the instruments used (Keithley 2400) did not support such continuous sweep.

**Data availability**. The datasets generated during and/or analysed during the current study are available from the corresponding author on reasonable request.

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

## Acknowledgements

This work was performed in part at the San Diego Nanotechnology Infrastructure (SDNI) of UCSD, a member of the National Nanotechnology Coordinated Infrastructure, which is supported by the National Science Foundation (grant ECCS-1542148). M.C.M.M.S. and N.C.F. acknowledge financial support from the São Paulo Research Foundation (2014/04748-2 and 2015/20525-6). M.C.M.M.S. acknowledges fruitful discussions with colleagues from Fainman's group at UCSD and from the Device Research Laboratory at Unicamp.

## Author contributions

A.G. and Y.F. conceived and supervised the project. M.C.M.M.S. and A.G. developed the mathematical model. N.C.F. co-supervised the project mostly with the mathematical model development. M.C.M.M.S. performed simulations, fabricated the samples, and realized the experiments. All authors contributed to discussing the results and preparing the manuscript.

## Additional information

**Competing interests:** The authors declare no competing financial interests.

