## [Peer Review File · Nature Communications]

Reviewers' comments:

Reviewer #1 (Remarks to the Author):

This manuscript introduces a Si-FTS based on waveguide MZI and micro-heater together with the associated math model which is developed to deal with the non-linearity, thermo-expansion and dispersion of the silicon waveguide and turn them into benefits. It is a nice and original piece of work and definitely deserves being published if the comments below are dealt with. However I feel that the degree of fundamental novelty of the presented work is not of a sufficient degree to motivate publication in NComms. Rather the paper describes a clever and sound piece of engineering, that should find its way into an appropriate photonics journal.

Below are a number of comments that may help to improve the quality of the manuscript:

1. In Figure 2a, the amplitude of the as-measured interferogram decreases as heater power increases. In Figure 2b, the envelope is observed to have small features and is more than just a simple fitting of the peaks. It would be useful if the authors can provide additional discussion regarding the origin of this amplitude decay and the details about the envelope. One important question is that whether the amplitude decay depends on wavelength or not. Since this will make a difference in the normalization of the interferogram generated by the broadband source.
2. Instead of mentioning that the calibration is done “for various laser frequencies”, it would be better to specify the number of frequencies that is necessary for the calibration.
3. While the authors claimed that the theoretical resolution of their device is improved to 0.38THz after non-linearity, thermo-expansion and dispersion is carefully accounted, the authors didn't include any experimental results regarding the resolution in the manuscript. It should be easy for the authors to provide some experimental results about resolution they can achieve experimentally.
4. Although a 7 THz wide spectrum is recovered as an example to demonstrate the broadband operation, there is no direct discussion about the bandwidth of the device in the manuscript. It should be useful if the authors can provide some comments on the wavelength range in which their device can operate. The shortest wavelength will be determined by the sampling interval of the interferogram. And the longest wavelength can be linked to the waveguide bend losses for example. Some information or comments would be helpful if readers want to compare the proposed device to other types of FTS.

5. The authors mentioned “the zero delay (maximum envelope amplitude) is shifted to around 17.2 ps ...”, but the shift of the zero delay is observed to be less than 0.2 ps in Figure 3c. In fact, if one put the numbers ($L = 30.407\text{mm}$, $c = 3 \cdot 10^8\text{m/s}$, $\delta(n_{\text{eff}}|_{(v_0)}) + v_0 \delta(\partial_v n) = 2 \cdot 10^{-3}$, $\xi_1 = 23 \cdot 10^{-2}$) into equation (16), one would get the zero delay of around 0.165 ps which matches what is observed in Figure 3c. It should be clarified if the “17.2 ps” is the value for another definition.
6. The expression followed after equation (16) does not match the expression in equation (16). It should be $\delta(n_{\text{eff}}|_{(v_0)}) + v_0 \delta(\partial_v n)$ rather than $\delta(n_{\text{eff}}|_{(v_0)}) + v_0 \delta(\partial_v n)$.
7. In the caption of Figure 3, “e-g” should be “d-f”.
8. There is an “.m” in “...agreement with the eq.14.m” that is not clear what is referred to.
9. In the Supplementary Info, the “0.2” in the sentence “In our experiments we used ... around 0.2” should be “0.02”.

Reviewer #2 (Remarks to the Author):

Viewing from the pdf file is preferred, as some symbols may not appear properly.

The paper presents a Fourier transform (FT) microspectrometer based on temperature-induced phase scans of a Mach-Zehnder Interferometer (MZI) implemented in silicon waveguides. A resolution of 3 nm in the 1.4 - 1.6 μm band is experimentally demonstrated. The temperature change is generated by resistive heating, consuming up to 12 mA current and 600 mW power. Despite the device structure being quite straightforward in FT spectrometry, the paper thoroughly discloses calibration and spectral retrieval methods which effectively suppress dispersive and non-linear thermal effects. This enables accurate operation of the device while using a broad temperature scan range, paving the way towards further resolution increases and device integration in mobile platforms.

The paper is well-written and the proposed techniques provide a significant contribution to the development of integrated spectrometers that, in my view, merit publication in nature communications. However, I believe that the claims by the authors regarding the state of the art, the device advantages and its scalability could mislead the reader and should be reformulated, as detailed below. Key performance parameters (resolution, free-spectral-range and power consumption) should be stated in the abstract and introduction. Some minor typos and issues

regarding the figures and their description should also be addressed.

1. Resolution and free spectral range

Resolution is one of the most important parameters that enable the reader to evaluate the performance of a spectrometer, yet it is only introduced in the final discussion that the resolution is about 3 nm. This resolution limits the type of spectral features that can be resolved and therefor the range of applications. The authors chose to retrieve the ASE spectrum of an EDFA, citing availability in the lab as one reason. Since the ASE spectrum only has broad features, this fact should be pointed out. I believe the resolution and the free spectral range (FSR) of the device should be mentioned both in the abstract and the introduction. A brief discussion of the inherent linkage between these two parameters would be appropriate.

Furthermore, it is said that traditional FTS research is scarce compared to other alternatives, and that the reasons are unknown. Since no performance data is provided in this introduction, I believe this statement disregards the fact that the same interferometer length provides much greater resolutions in other schemes such as SHS. For example in the references listed below, in [1] a resolution of 1 pm is presented in a simulated device, whereas in [2] a similar MZI arm length of 3 cm results in an experimental resolution of 17 pm (several orders of magnitude greater than in this paper). Of course, the FSR of these devices is extremely narrow and their application and challenges are different, so they do not reduce the merit of this contribution. However, I believe that a comprehensive state-of-the-art overview should include this discussion, cite the relevant references, and place the different types of FTS in proper context.

Finally, it should be mentioned that micro-heaters have already been used in the context of silicon-based FT spectrometers [3], even though they were employed for phase correction in SHS and not for the full temperature sweep presented in this paper. The associated power consumption of the current device should be mentioned in the abstract and introduction.

[1] (Already cited) B. Imran Akca, "Design of a compact and ultrahigh-resolution Fourier-transform spectrometer," *Opt. Express* 25, 1487-1494 (2017)

[2] Alaine Herrero-Bermello, Aitor V. Velasco, Hugh Podmore, Pavel Cheben, Jens H. Schmid, Siegfried Janz, María L. Calvo, Dan-Xia Xu, Alan Scott, and Pedro Corredera, "Temperature dependence mitigation in stationary Fourier-transform on-chip spectrometers," *Opt. Lett.* 42, 2239-2242 (2017)

[3] K. Okamoto, H. Aoyagi, and K. Takada, "Fabrication of Fourier-transform, integrated-optic spatial heterodyne spectrometer on silica-based planar waveguide," *Opt. Lett.* 35, 2103-2105 (2010).

2. Scalability

The authors claim that this scheme could be scaled to a resolution of 0.7 pm using the 64-cm low-loss waveguides in reference 32. I believe this specific claim to be too bold and too speculative, since despite their low losses, the accumulated loss in the 64 cm path is still 17.5 dB, and its shallow ridge structure may present a different thermo-optic profile. The scalability is definitely a great advantage of the device, but I believe that particular claim is unsubstantiated and unnecessary.

To this regard, I believe the propagation losses of the MZI waveguides, as well as the bandwidth and coupling losses of the input coupler should be included in the text, as they provide more straight-forward information about the scalability of the device in terms of resolution and FSR.

3. Non-linearities as an advantage, power consumption

Both in the abstract and the discussion (page 5, paragraph 2), it is stated that the deviations due to the dispersive and non-linear effects enhance the device resolution. It's only on the final page it states that the comparison is for a given dissipated power. I believe this is misleading, since there cannot be a device without these effects to compare to (i.e. δv_{ideal} is artificial). In fact, these effects must be accounted for, otherwise the spectra would be extremely poorly retrieved as very clearly shown by figure 3d. I believe the fact that the proposed corrections enable operation in a previously inoperable temperature range is significant enough, and that both the claim in the abstract and the corresponding discussion should be removed.

Additionally, the spectrometer consumes significant power (operating with up to 12 mA and 600 mW). Another way to look at the above points is that utilizing the TOC non-linearity and dispersion makes the heating more efficient in inducing required phase change. Modifying the waveguide cross-section, spacer thickness and adding heat insulating trenches are all possibilities to further increase the heating efficiency, albeit with their corresponding compromises as well. I believe presenting the use of TOC non-linearity and dispersion as means of reducing power consumption may be more appropriate.

4. Figures

Finally some minor issues regarding typos and figures.

-There are a number of small grammar errors and typos that should be carefully edited. For example, the thickness of the buried oxide is 3 μm (not 3 mm, p.5, Fabrication).

-It should be mentioned in the text how the envelope of Fig. 2b is obtained.

-The caption of Fig.2 includes some explanatory sentences that are insufficient to fully

understand the process by themselves, and which are already properly explained in the main text. I think removing them and leaving the short description of the panel content would improve clarity.

-The small size of the insets in Fig 3c makes it difficult to appreciate the discussed phase changes. I would suggest depicting fewer periods to make it more noticeable.

Response to reviewers' comments

(Manuscript NCOMMS-17-26038)

In the following, the reviewers' comments are reproduced and the authors' point-by-point replies are provided in blue. In addition, modifications in the manuscript text are highlighted here in red.

Reviewer #1 (Remarks to the Author):

This manuscript introduces a Si-FTS based on waveguide MZI and micro-heater together with the associated math model which is developed to deal with the non-linearity, thermo-expansion and dispersion of the silicon waveguide and turn them into benefits.

It is a nice and original piece of work and definitely deserves being published if the comments below are dealt with. However I feel that the degree of fundamental novelty of the presented work is not of a sufficient degree to motivate publication in NComms. Rather the paper describes a clever and sound piece of engineering that should find its way into an appropriate photonics journal.

Author's response (A.R.): Although it may not provide fundamental novelty in terms of device concept, motivated by its novel mobile application, our thorough discussion of the Si-FTS, including calibration and spectral retrieval methods, is of significant practical interest for integrated photonics spectroscopy and provides a clear path for realizing such devices in mobile platforms.

As such, we believe it would be suitable for publication in Nature Communications as stated in the journal policy: "While Nature and the Nature research journals strive to publish the highest impact research within these disciplines, **Nature Communications is committed to publishing important advances of significance to specialists within each field.**" (<https://www.nature.com/ncomms/journal-policies/guide-to-referees>)

Below are a number of comments that may help to improve the quality of the manuscript:

1. In Figure 2a, the amplitude of the as-measured interferogram decreases as heater power increases. In Figure 2b, the envelope is observed to have small features and is more than just a simple fitting of the peaks. It would be useful if the authors can provide additional discussion regarding the origin of this amplitude decay and the details about the envelope. One important question is that whether the amplitude decay depends on wavelength or not. Since this will make a difference in the normalization of the interferogram generated by the broadband source.

A.R.: Neither the amplitude decay nor the features in the envelope depend on wavelength. They are caused, respectively, by slight misalignment and vibration of the input/output fibers as the heater power increases due to the thermal expansion of the chip. In our experiments the fibers were not permanently attached to the chip, as described in the Methods section:

"Light was coupled into and out of the device using [...] cleaved fibers. (...) The fibers made physical contact with the sample, but were not permanently attached to it."

We introduced the following extract in the text do address the issue raised by the reviewer:

"The decrease in the mean optical power in Fig.2a and the small features in the envelope of Fig.2b are caused by slight misalignment and vibration of the input/output fibers as the heater power increases due to the thermal expansion of the chip. In a practical packaged application, these features would not be present as the fiber would be permanently attached to the silicon chip."

2. Instead of mentioning that the calibration is done “for various laser frequencies”, it would be better to specify the number of frequencies that is necessary for the calibration.

A.R.: The calibration involves nonlinear fitting of the interferogram $I(W)$ at various frequencies to obtain $K(\nu)$ and γ_w , followed by a linear regression of $K(\nu)$. While three frequency points (as we used) suffice to obtain the linear regression, a larger number of points would decrease the statistical error.

Notice, however, in our demonstration the total error is dominated by the error bars of each data point obtained from the non-linear fit of $I(W)$, not by the statistical error of the linear regression. Thus, increasing the number of frequency points would have little effect on the calibration.

We slightly changed the phrasing in the manuscript to: “The calibration [...] is achieved measuring the interferogram of the laser source at **various different laser frequencies (at least three) in the spectral region of interest.**”

3. While the authors claimed that the theoretical resolution of their device is improved to 0.38THz after non-linearity, thermo-expansion and dispersion is carefully accounted, the authors didn't include any experimental results regarding the resolution in the manuscript. It should be easy for the authors to provide some experimental results about resolution they can achieve experimentally.

A.R.: The resolution of 0.38 THz is in fact the *experimental* resolution. To clarify this issue, we introduced the word “experimental” explicitly in the text to facilitate interpretation:

“(...) The **experimental** spectral resolution of $\delta\nu= 0.38$ THz ($\delta\sigma= 12.7$ cm⁻¹, $\delta\lambda = 3.05$ nm) is comparable to other on-chip spectrometers aimed at broadband operation (...)”.

4. Although a 7 THz wide spectrum is recovered as an example to demonstrate the broadband operation, there is no direct discussion about the bandwidth of the device in the manuscript. It should be useful if the authors can provide some comments on the wavelength range in which their device can operate. The shortest wavelength will be determined by the sampling interval of the interferogram. And the longest wavelength can be linked to the waveguide bend losses for example. Some information or comments would be helpful if readers want to compare the proposed device to other types of FTS.

A.R.: The reviewer is correct in stating that the shortest wavelength will be determined by the sampling interval of the interferogram. In practice, this does not pose a limitation since fine sampling can be easily obtained with thermal heaters.

What limits the bandwidth of the Si-FTS in a real device is the finite bandwidth of the waveguide power splitter/combiner.

Regarding our particular device, the non-flat MZI transmission of Fig.2h is attributed to the non-ideal response of the power splitter for frequencies higher than 194 THz. We added the following sentence to address this point in the manuscript:

“The non-flat transmission in Fig.2h is attributed to the non-ideal response of the power splitter/combiner for frequencies higher than 194 THz. In practice, the limited bandwidth of these components will dictate the bandwidth of the Si-FTS. For the specific y-junction design used here³⁰, operation over more than 25 THz (200 nm), excess loss lower than 0.3 dB and low reflection is expected.”

A general comment on this respect is also present further in the manuscript, in the Discussion section:

“The ultimate performance of the on-chip Si-FTS is quite promising considering recent advancements in silicon photonics design and fabrication. **First, the window of operation for a given device will be dictated by the finite bandwidth of the waveguide optical power couplers/splitters.** ~~Power couplers/splitters~~ Components offering flat optical response over tens of terahertz³⁰⁻³² (hundreds of nanometers) and extremely low excess loss may allow Si-FTS operating over large bandwidths.”

5. The authors mentioned “the zero delay (maximum envelope amplitude) is shifted to around 17.2 ps ...”, but the shift of the zero delay is observed to be less than 0.2 ps in Figure 3c. In fact, if one put the numbers ($L = 30.407\text{mm}$, $c = 3 \cdot 10^8\text{m/s}$, $\delta(n_{\text{eff}}|_{v_0}) + v_0 \delta(\partial_v n) = 2 \cdot 10^{-3}$, $\xi_1 = 23 \cdot 10^{-2}$) into equation (16), one would get the zero delay of around 0.165 ps which matches what is observed in Figure 3c. It should be clarified if the “17.2 ps” is the value for another definition.

A.R.: It was our error. The correct value is 0.172 ps instead of 17.2 ps. This has been corrected in the text. Using 0.172 ps in eq.16 we get

$$\delta(n_{\text{eff}}|_{v_0}) + v_0 \delta(\partial_v n) \cong 2.1 \cdot 10^{-3}.$$

6. The expression followed after equation (16) does not match the expression in equation (16). It should be $\delta(n_{\text{eff}}|_{v_0}) + v_0 d(\partial_v n)$ rather than $n_{\text{eff}}|_{v_0} + v_0 d(\partial_v n)$.

A.R.: Typo corrected.

7. In the caption of Figure 3, “e-g” should be “d-f”.

A.R.: Typo corrected.

8. There is an “.m” in “...agreement with the eq.14.m” that is not clear what is referred to.

A.R.: “m” removed.

9. In the Supplementary Info, the “0.2” in the sentence “In our experiments we used ... around 0.2” should be “0.02”.

A.R.: Typo corrected.

Reviewer #2 (Remarks to the Author):

The paper presents a Fourier transform (FT) micro-spectrometer based on temperature-induced phase scans of a Mach-Zehnder Interferometer (MZI) implemented in silicon waveguides. A resolution of 3 nm in the 1.4 - 1.6 μm band is experimentally demonstrated. The temperature change is generated by resistive heating, consuming up to 12 mA current and 600 mW power. Despite the device structure being quite straightforward in FT spectrometry, the paper thoroughly discloses calibration and spectral retrieval methods which effectively suppress dispersive and non-linear thermal effects. This enables accurate operation of the device while using a broad temperature scan range, paving the way towards further resolution increases and device integration in mobile platforms.

The paper is well-written and the proposed techniques provide a significant contribution to the development of integrated spectrometers that, in my view, merit publication in nature communications. However, I believe that the claims by the authors regarding the state of the art, the device advantages and its scalability could mislead the reader and should be reformulated, as detailed below. Key performance parameters (resolution, free-spectral-range and power consumption) should be stated in the abstract and introduction. Some minor typos and issues regarding the figures and their description should also be addressed.

1. Resolution and free spectral range

- 1.1. Resolution is one of the most important parameters that enable the reader to evaluate the performance of a spectrometer, yet it is only introduced in the final discussion that the resolution is about 3 nm. This resolution limits the type of spectral features that can be resolved and therefore the range of applications. The authors chose to retrieve the ASE spectrum of an EDFA, citing availability in the lab as one reason. Since the ASE spectrum only has broad features, this fact should be pointed out.

A.R.: We introduced a sentence in response to the reviewer's comment:

"The ASE provides a good test spectrum in the telecom band, suitable for testing with the available equipment in our lab. Also, the broad features of the ASE spectrum are suitable for this demonstration given the limited resolution achieved here (0.38 THz)."

- 1.2. I believe the resolution and the free spectral range (FSR) of the device should be mentioned both in the abstract and the introduction. A brief discussion of the inherent linkage between these two parameters would be appropriate.

A.R.: Differently from designs such as SHS, the FT spectrometer with true time delay does not have a fixed FSR that is related to the resolution. Instead, the resolution is directly dictated by the maximum achievable arm delay $\Delta\tau$, which in turn is dictated by the arm length and power consumption.

Information on the resolution of the device, bandwidth of the light source and power dissipation has been introduced in the abstract:

"We calibrate the Si-FTS, including the correction parameters, using a tunable laser source and we successfully retrieve the PSD of a broadband source (7 THz around 193.4 THz) with a resolution of 0.38 THz and total electric power consumption of 2.5 W per heater. ~~The aforementioned effects are shown to effectively enhance the Si-FTS resolution when properly accounted for~~"

We also introduced this information in the introduction along with the other results of the paper:

"In this article, we demonstrate the implementation of a Si-FTS on the SOI platform with integrated microheaters. We show that the issues related to thermo-optic non-linearity, thermal expansion and dispersion can be properly understood ~~tackled and, moreover, ultimately result in enhanced performance~~ and incorporated in a simple manner. We derive a FT relation between PSD and interferogram with modified optical frequency and arm delay accounting for these effects and we demonstrate a calibration procedure using a tunable laser source. Further, we demonstrate the retrieval of a 7-THz-wide light source around 193.4 THz with spectral resolution of 0.38 THz (12.7 cm^{-1} , 3.05 nm) using a 1 mm² device with total electric power dissipation around 2.5 W per heater. The Si-FTS shows intrinsic resilience to fabrication variations that allows scalability of its resolution and power consumption performance, enabling robust and versatile portable spectrometers.

- 1.3.** Furthermore, it is said that traditional FTS research is scarce compared to other alternatives, and that the reasons are unknown. Since no performance data is provided in this introduction, I believe this statement disregards the fact that the same interferometer length provides much greater resolutions in other schemes such as SHS. For example in the references listed below, in [1] a resolution of 1 pm is presented in a simulated device, whereas in [2] a similar MZI arm length of 3 cm results in an experimental resolution of 17 pm (several orders of magnitude greater than in this paper). Of course, the FSR of these devices is extremely narrow and their application and challenges are different, so they do not reduce the merit of this contribution. However, I believe that a comprehensive state-of-the-art overview should include this discussion, cite the relevant references, and place the different types of FTS in proper context.

[1] (Already cited) B. Imran Akca, "Design of a compact and ultrahigh-resolution Fourier-transform spectrometer," *Opt. Express* 25, 1487-1494 (2017).

[2] Alaine Herrero-Bermello, Aitor V. Velasco, Hugh Podmore, Pavel Cheben, Jens H. Schmid, Siegfried Janz, María L. Calvo, Dan-Xia Xu, Alan Scott, and Pedro Corredera, "Temperature dependence mitigation in stationary Fourier-transform on-chip spectrometers," *Opt. Lett.* 42, 2239-2242 (2017).

A.R.: As pointed out by the reviewer, FT-based spectrometers such as SHS are suitable for high-resolution, narrow bandwidth applications, whereas the traditional FTS design such as the one discussed in this work is more suitable for moderate resolution, broad bandwidth applications.

We believe that the remark on the scarcity of research on the traditional FTS does not demean the impressive results achieved with SHS precisely because these two spectrometer designs address different needs.

We introduced a sentence in the introduction to make this distinction clearer for the reader:

"Investigations of the traditional FTS design in the silicon photonics platform have been, however, surprisingly scarce²⁹. Whereas FT-based spectrometers such as SHS are suitable for high-resolution, narrowband applications, the traditional FTS is a promising candidate to address moderate resolution, broadband applications. Although one can only speculate about the reasons for such scarcity, some challenges can be identified when considering the requirements for a silicon photonics-based FTS (Si-FTS)."

Also, we added reference [2] to the section mentioning SHS.

- 1.4.** Finally, it should be mentioned that micro-heaters have already been used in the context of silicon-based FT spectrometers [3], even though they were employed for phase correction in SHS and not for the full temperature sweep presented in this paper. The associated power consumption of the current device should be mentioned in the abstract and introduction.

[3] K. Okamoto, H. Aoyagi, and K. Takada, "Fabrication of Fourier-transform, integrated-optic spatial heterodyne spectrometer on silica-based planar waveguide," *Opt. Lett.* 35, 2103-2105 (2010).

A.R.: Information on the power consumption was introduced both in the abstract and in the introduction, as shown in the reply to item 1.2.

In the aforementioned reference, thermal tuning is used to compensate for phase errors on the various MZIs of a SHS. Given the differences in device operation, we believe that analyzing this example in the manuscript would distract from the discussion.

2. Scalability

The authors claim that this scheme could be scaled to a resolution of 0.7 pm using the 64-cm low-loss waveguides in reference 32. I believe this specific claim to be **too bold and too speculative**, since despite their low losses, the accumulated loss in the 64 cm path is **still 17.5 dB**, and its shallow ridge structure may present a **different thermo-optic profile**. The scalability is definitely a great advantage of the device, but I believe that particular claim is unsubstantiated and unnecessary.

A.R.: We agree with the reviewer's opinion that such specific claim may sound speculative. It has been removed from the discussion.

3. To this regard, I believe the propagation losses of the MZI waveguides, as well as the bandwidth and coupling losses of the input coupler should be included in the text, as they provide more straight-forward information about the scalability of the device in terms of resolution and FSR.

A.R.: We introduced the sentence "The propagation losses of the waveguides are estimated to be around 2 dB/cm" in Experimental Device subsection.

Regarding the input/output couplers, in the absence of direct measurements of the aforementioned parameters we provided some information on their expected performance: "For the specific y-junction design used here³⁰, operation over more than 25 THz (200 nm), excess loss lower than 0.3 dB and low reflection is expected."

4. Non-linearities as an advantage, power consumption

4.1. Both in the abstract and the discussion (page 5, paragraph 2), it is stated that the deviations due to the dispersive and non-linear effects enhance the device resolution. It's only on the final page it states that the comparison is for a given dissipated power. I believe this is misleading, since there cannot be a device without these effects to compare to (i.e $\delta\nu_{ideal}$ is artificial). In fact, these effects must be accounted for, otherwise the spectra would be extremely poorly retrieved as very clearly shown by figure 3d. I believe the fact that the proposed corrections enable operation in a previously inoperable temperature range is significant enough, and that both the claim in the abstract and the corresponding discussion should be removed. Additionally, the spectrometer consumes significant power (operating with up to 12 mA and 600 mW). Another way to look at the above points is that utilizing the TOC non-linearity and dispersion makes the heating more efficient in inducing required phase change. Modifying the waveguide cross-section, spacer thickness and adding heat insulating trenches are all possibilities to further increase the heating efficiency, albeit with their corresponding compromises as well. I believe presenting the use of TOC non-linearity and dispersion as means of reducing power consumption may be more appropriate.

A.R.: In order to prevent confusion, we followed the reviewer's suggestion to avoid stating that the resolution is improved by dispersion and thermo-optic nonlinearity. Such claims have been removed from the abstract and the introduction.

In the Discussion section, we modified the discussion to make it clear the point raised by the reviewer regarding the interplay between resolution and power consumption. The new version is as follows:

It is worth noticing that the presence of TOC non-linearity and dispersion have an upside in the Si-FTS performance. Ideally, one seeks to low the *resolution × dissipated power* product, given by

$$\delta\nu \times W_{total} = \kappa_T^{-1}$$

in the case where the two effects are absent (κ_T is assumed to be identical for the two heaters). In the presence of ξ_1 and γ_2 , this product is modified to

$$\delta\nu \times W_{total} = \kappa_T^{-1} [(1 + \xi_1)(1 + \gamma_2 \kappa_T W_{total})]^{-1}$$

and is effectively reduced, resulting in decreased power dissipation for a given resolution or, conversely, lower $\delta\nu$ given a maximum power. In our case, the resolution of 0.38 THz is achieved dissipating a total power of 5.1 W, representing a 35% decrease in the power (6.9 W) that would be required to achieve this resolution if dispersion and thermo-optic non-linearity were absent.

5. Figures

Finally some minor issues regarding typos and figures.

- 5.1. There are a number of small grammar errors and typos that should be carefully edited. For example, the thickness of the buried oxide is 3 μm (not 3 mm, p.5, Fabrication).

A.R.: We thoroughly reviewed the text for typos and mistakes.

- 5.2. It should be mentioned in the text how the envelope of Fig. 2b is obtained.

A.R.: We introduced a sentence in the caption of Fig.2 explaining how the envelope is calculated: “The envelope in **b** is the absolute value of the interferogram's Hilbert transform.”

- 5.3. The caption of Fig.2 includes some explanatory sentences that are insufficient to fully understand the process by themselves, and which are already properly explained in the main text. I think removing them and leaving the short description of the panel content would improve clarity.

A.R.: We reformulated the first part of the caption to improve clarity. The original version reads:

“(…) **a-f**. data in blue and red are related to heaters H1 and H2, respectively. **a**. As-measured interferogram at 183.37 THz (1600 nm) as a function of the heating power (W). **b**. To achieve robust non-linear fitting we subtract the mean power (red trace) and the envelope (red trace in **b**) to obtain the curve in **c**. **c**. The fitted curve (dashed-red trace) using eq.13 adjusts well the experimental trace. (…)”

The modified version reads:

“(…) **a-c**. Interferogram at 183.37 THz (1600 nm) as a function of dissipated power in heater H₂. The mean power (red trace in **a**) and the envelope (red trace in **b**) are subtracted to obtain the curve in **c**, fitted (dashed-red trace) using eq.13. The envelope in **b** is the absolute value of the interferogram's Hilbert transform. **d-f**: data in blue and red are related to heaters H1 and H2, respectively. (…)”

In addition, we removed the sentence “The linearity of the IV curves confirm the thermo-optic origin of the non-linear behavior observed in the interferogram” from the caption of Fig.2f.

- 5.4. The small size of the insets in Fig 3c makes it difficult to appreciate the discussed phase changes. I would suggest depicting fewer periods to make it more noticeable.

A.R.: The number of periods was reduced in both insets to facilitate visualization.

Reviewers' Comments:

Reviewer #1 (Remarks to the Author):

The authors have addressed the comments from the reviewers well. Various problems and errors have been dealt with.

In that sense the paper appears to be ready for publication.

But I continue to believe that the presented work is not of a degree of novelty that justifies publication in Nature Photonics. In essence the paper reports about a very conventional FTIR configuration translated to PIC technology. The challenges involved in this are in essence engineering problems.

Reviewer #2 (Remarks to the Author):

The authors have adequately addressed my concerns. I now recommend the publication of this manuscript.